# Molecular Insights into Host and Vector Manipulation by Plant Viruses

**DOI:** 10.3390/v12030263

**Published:** 2020-02-27

**Authors:** Véronique Ziegler-Graff

**Affiliations:** Institut de Biologie Moléculaire des Plantes, CNRS, Université de Strasbourg, 67084 Strasbourg, France; veronique.ziegler-graff@ibmp-cnrs.unistra.fr

**Keywords:** virus, vector, plant host, transmission, manipulation, defence mechanisms, pathogenicity factor, suppressor of RNA silencing

## Abstract

Plant viruses rely on both host plant and vectors for a successful infection. Essentially to simplify studies, transmission has been considered for decades as an interaction between two partners, virus and vector. This interaction has gained a third partner, the host plant, to establish a tripartite pathosystem in which the players can react with each other directly or indirectly through changes induced in/by the third partner. For instance, viruses can alter the plant metabolism or plant immune defence pathways to modify vector’s attraction, settling or feeding, in a way that can be conducive for virus propagation. Such changes in the plant physiology can also become favourable to the vector, establishing a mutualistic relationship. This review focuses on the recent molecular data on the interplay between viral and plant factors that provide some important clues to understand how viruses manipulate both the host plants and vectors in order to improve transmission conditions and thus ensuring their survival.

## 1. Introduction

The vast majority of plant viruses that are economically important in agriculture are transmitted by vectors [1]. Insects, especially whiteflies and aphids, transmit more than 50% of all vectored-viruses. Two main modes of transmission by vectors are described: persistent or non-persistent transmission, depending on the time window required by vectors to acquire and further disseminate the virus [2,3]. Non-persistently transmitted viruses are retained only for seconds to minutes in the insect stylets or foregut and rapidly released during salivation; internalisation into the vector body does not occur. Persistent (also called circulative) viruses have to be taken up through the digestive tract and enter the insect body where they traffic in the haemolymph to reach the salivary glands from which they are injected with the saliva into new host plants. The virus can remain in the insect for a very long period up to the end of its life [1,4]. A small number of virus species also replicate within the cells of the vector and this transmission mode is therefore called circulative propagative [3]. Whatever the virus retention site or the journey through the insect, transmission relies on specific interactions between viral proteins and vector proteins [5]. Viral determinants are particularly well characterized [2,4,6] while their vector counterparts, in particular the receptor cuticular proteins, are only beginning to be characterized [7].

Plant viruses induce during infection numerous changes in the physiology of the host during infection which can influence the vector’s behaviour and/or performance, and which can, as a consequence, modify viral propagation, a topic covered by many recent reviews [8,9,10,11,12,13]. The manipulative effects of plant viruses on their vectors operate at several levels: modified perception of visual and volatile sensory cues, feeding behaviour (depending on the nutritious quality of the plant) and fecundity [14]. These virus-induced changes occur indirectly through the host plant and play a role in most vectored virus transmissions. However, some viruses can also alter the vector behaviour directly, which means the modifications occur within the vector itself after it has acquired virus [13]. This is evidently the case for persistently transmitted plant viruses that replicate within insect vectors [15]. A few examples of circulative non replicative viruses have also been documented. For instance, aphids carrying luteovirids (viruses of the *Luteoviridae* family) showed preference for non-infected plants while non viruliferous aphids were attracted by volatiles emitted by infected plants [16,17]. The interactions between viruses and vectors can range from detrimental (due to the poor quality or even toxicity of the infected plant) to beneficial (the vector performances are improved). In this latter case, the relationship can be considered as mutualistic and both partners proliferate to a greater extent than being alone [12].

Recent interest has focused on a limited number of pathosystems for which the molecular mechanisms set up in infected plants by specific viral proteins have been studied. They induce changes that are beneficial or detrimental to insects and as a consequence, to the virus itself when dissemination to other plants and survival is considered. Some of these changes modify the visual and olfactory cues of the plant to attract or promote feeding of the insect. More often a distinct viral factor will target the plant defence mechanisms triggered by the vector, which will thus be attenuated and favour the insect attraction, settlement and performance and, as a consequence, foster the viral [10,18]. This review summarises the latest findings on the viral gene products of two non-persistently and three persistently transmitted viruses belonging to some of the most economically important viruses pathogenic to crops. The discussed proteins were often initially identified as pathogenicity factors or suppressors of gene silencing, and the interacting plant factors or pathways, mainly as defence-related pathways. Some lines of future research are suggested to further unravel this particularly exciting field of interactions between viruses, plant hosts and vectors.

## 2. The Interplay between CMV 2b and JA Signalling

*Cucumber mosaic virus* (CMV, *Bromoviridae* family) is a non-persistent virus transmitted by over 80 species of aphids and it has an extremely broad host range [19,20]. It causes a serious disease in agriculture. Retention of CMV particles to the stylets of the aphid vector and release from them occur during brief test punctures on infected and healthy plants, respectively, and is mediated by the interaction of specific amino acids of the coat protein with unknown aphid receptors [4]. A large body of evidence has accumulated showing that CMV-infected plants are generally more attractive to aphids (reviewed in [9,10,11]). However, in a few cases, opposite effects on host-aphid interactions were observed, depending on the combination of the three partners including the viral strain [12]. The seminal work by Mauck et al. [21] on this non-persistent model showed that the preference of two generalist aphid species for Fny-CMV (Fny strain)-infected squash plants was related to higher emissions of volatile organic compounds (VOC) released by these plants. However, the poor nutritious quality of the infected plants incited the aphids to emigrate to new plants. A model conducive for non-persistent transmitted viruses, also called “push-pull strategy”, was proposed in which an initial increase in aphid attractiveness by deceptive volatiles is followed by feeding deterrence by plants distasteful to the insects, a situation favouring rapid dissemination of these viruses that are acquired during brief aphid punctures in the epidermis and mesophyll of infected leaves [22]. 

Most reports on molecular mechanisms that rule plant and aphid changes induced by CMV infection focus on the 2b protein. 2b is the RNA silencing suppressor (RSS) of CMV. RNA silencing (or RNA interference, RNAi) is a highly conserved mechanism that inactivates gene expression to regulate a wide range of biological processes including antiviral defence [23,24]. Viruses are both inducers and targets of RNA silencing. RNAi-mediated antiviral plant immunity is triggered by double-stranded RNA (dsRNA) derived from replication intermediates of RNA viruses, secondary RNA structure or bidirectional transcription of DNA viruses. dsRNA is processed by RNAse III-type enzyme Dicer-like proteins (DCL) into viral small RNA duplexes (viRNA). The viRNAs are then loaded into ARGONAUTE (AGO) proteins, the main components of the RNA Induced Silencing Complex (RISC) that is able to cleave the target RNA (e.g., the viral RNA) sequence specifically. To counteract this defence mechanism viruses have developed proteins called RNA silencing suppressors (RSS) that interfere with the RNAi pathway at virtually all steps. The case of the CMV 2b protein is remarkable. Despite its small size (11.5 kDa) several modes of action and targets have been reported (reviewed in [25,26]) such as dsRNA binding to prevent the long distance spread of signalling signals, interacting with AGO1 to inhibit RISC slicer activity, and with AGO4 to modulate endogenous transcription.

In a series of articles, the group of Carr investigated the role of 2b in plant defence signalling and its link to aphid transmission. By a transcriptional approach using transgenic arabidopsis plants constitutively expressing the 2b protein (Fny-CMV strain) they found that the viral protein repressed the expression of 90% of the genes regulated by jasmonic acid (JA), but not those involved in JA biosynthesis [27]. Conversely, a number of genes normally unaffected by salicylic acid (SA) became responsive. As JA is an important regulator in insect resistance, including aphids and whiteflies [28], Lewsey et al. [27] hypothesized that the 2b protein alters the hormonal balance in CMV-infected plants and thereby generates favourable conditions to promote aphid-mediated virus transmission. This hypothesis was confirmed by Ziebell et al. [29] who showed that a CMV∆2b mutant which does not express the 2b protein, elicits strong resistance to aphids in tobacco plants. 

More recently, Wu et al. [30] explored the molecular strategy developed by the 2b protein (SD-CMV strain) to repress JA-signalling. They showed that like in the former experiments on squash [21] aphid vectors were attracted by odours emitted by CMV-infected arabidopsis. Attraction was specifically correlated to the presence of 2b, as shown both with transgenic plants expressing the 2b protein and with CMV-infected plants, compared to plants infected with CMV∆2b. Wu et al. [30] further demonstrated that the 2b protein specifically and directly interacted with several JAZ proteins (in particular with JAZ1). JAZ proteins are the key repressors of JA signalling [31,32]. Under controlled conditions, the low JA level favours the accumulation of JAZ proteins which bind to and repress the activity of transcription factors like MYC2 that positively regulate JA-responsive genes (Figure 1, panel A). Under stress conditions, JA level is highly increased and JA-specific degradation of JAZ proteins occurs via the 26S proteasome releasing the MYC factors which can activate the JA responses (Figure 1, panel B). When 2b binds to JAZ proteins, the JAZ degradation by the E3 ubiquitin ligase SCF^COI1^ is repressed and JA signalling is inhibited. The higher attraction of aphids by plants expressing the viral 2b protein is recapitulated on arabidopsis mutants defective in JA perception (*coi1-1)* or in downstream JA-regulated genes (*myc2/3/4*) confirming that 2b represses JA-mediated signalling. This action is independent of its RSS activity as shown by the functionality of a 2b mutant lacking the RSS activity. Finally, the fact that the 2b proteins of different CMV host strains interact with JAZ proteins suggests an evolutionary conserved mechanism.

As mentioned before, the outcome of an infection resulting from plant-aphid interactions can vary in different hosts and this can be elicited by different viral proteins. In contrast to the results on cucurbit plants [21] where Fny-CMV-infected plants emitted deceptive aphid-attracting volatiles and were of low nutrition quality, tobacco plants infected with the same strain did not modify the settling of aphids and were of higher quality for aphid survival and reproduction [33], a clear mutualistic situation. However, tobacco plants infected with CMV∆2b showed a dramatic reduction in aphid populations, indicating that the 2b protein was able to repress resistance to aphid infestation in tobacco. By using viral reassortants created by shuffling the genomic RNAs from Fny-CMV and LS-CMV strains, Tungadi et al. [34] showed that in tobacco, Fny-CMV but not LS-CMV RNA 1 (or its 1a protein, a replication-associated factor) triggers resistance to aphids, while the 2b protein of both strains suppressed aphid resistance. In contrast, in arabidopsis aphid antibiosis is induced by Fny-CMV 2b protein and 1a prevents 2b’s effect [35]. This shows that in the two host plants 1a and 2b proteins play antagonistic roles. The complexity of the interactions between the CMV proteins and the host factors was further exemplified [35]: in arabidopsis Fny-CMV induces the biosynthesis of the glucosinolate 4-methoxy-indol-3-yl-methylglucosinolate (4MI3M), an aphid deterrent molecule [36]. The synthesis of 4MI3M is under positive control of AGO1, which can be degraded by the RSS activity of 2b. This suggests that 2b can negatively regulate the accumulation of the toxic glucosinolate. An additional layer of regulation by the viral proteins 1a and 2a (the RNA-dependant RNA polymerase) has been proposed [35]. These studies, performed on one single pathosystem with two virus strains and different plant hosts, highlight the complexity of the molecular interplay that drives the interactions between virus, plant host and vector. Such diversity in host adaptation, reported in several reviews [8,11,12], may become highly beneficial when aphids have to adapt to seasonal changes or host availability [35].

## 3. Potyviruses: An Intriguing Strategy of Protein Relocalisation to Manipulate Aphids

Potyviruses are one of the most widely distributed groups of plant viruses and many of them are efficiently transmitted by aphid vectors in a non-persistent manner [2,37]. Viral particles are transported to new plants by being transiently attached to the aphid’s stylets [4]. Former studies showed that aphids often prefer to settle on potyvirus-infected plants and reproduce better on them compared to control plants [9].

Casteel and collaborators conducted pioneer work to understand the mechanism underlying the potential synergism between *Turnip mosaic virus* (TuMV) and the aphid vector, *Myzus persicae*. In a first study, they transiently over-expressed ten individual TuMV proteins in Agrobacterium-infiltrated *Nicotiana benthamiana* leaves and recorded their effect on aphid reproduction [38]. A single protein, the nuclear inclusion a-protease domain NIa-Pro, was found to significantly improve *M. persicae*’s growth and settling preference of apterous aphids. NIa-Pro is the main protease of TuMV cleaving the viral polyprotein into individual proteins. A non-specific DNAse activity has been demonstrated for the protein encoded by two other potyviruses [39,40], but no silencing suppression activity has yet been reported for NIa-Pro. The effect on aphid’s fecundity was reproduced on transgenic arabidopsis plants stably expressing the viral protein. Furthermore NIa-Pro-expressing leaves showed a significant reduction of callose induction, a well-characterized plant response to aphid feeding, suggesting a role of NIa-Pro in reducing plant defences. They further investigated the sub-localisation of NIa-Pro and could correlate the improvement of aphid performance to the targeting of the protein to the vacuole [41]. Using the same approach of transient expression of the GFP:NIa-Pro fusion protein in *N. benthamiana*, they showed that the presence of aphids triggered the relocalisation of the protein from the nucleus and cytoplasm to the vacuole, and that targeting is reversible. The high number of cells responding to aphids with vacuolar NIa-Pro suggested the existence of a systemic signal moving throughout the leaf to modify NIa-Pro’s localisation. This observation resembles a similar signalling event called transmission activation (TA) discovered in plants infected by *Cauliflower mosaic virus*, a virus transmitted by aphids in semi-persistent way [42]. TA occurs within seconds and is characterised by aphid-induced formation of transmissible viral complexes at the very early time points of aphid perception in viral-infected plants. Recently, a similar phenomenon has also been found with TuMV infections [43]. By using chemical drugs, Drucker and collaborators showed that TA depends on calcium and reactive oxygen species signalling for both viruses [44]. 

Importantly, when GFP:NIa-Pro was expressed from the TuMV genome, the same relocalisation of the fusion protein to the vacuole was correlated to the aphid’s presence. NIa-Pro encoded by Potato virus Y, another aphid-transmitted potyvirus, behaved similarly suggesting that the vacuolar localisation is a conserved feature among members of the Potyvirus genus. Both plant-species specific and vector-specific factors seem to be involved as no effect was observed when the protein was transiently expressed in *Nicotiana tabacum,* and neither whiteflies nor leafhoppers were able to induce the strong relocalisation response of NIa-Pro. These observations suggest that NIa-Pro vacuolar targeting might be an important response to aphid’s presence aimed to inhibit aphid-induced plant defence signals potentially stored in the vacuole.

Another line of research of Casteel’s group focused on the role of phytohormones in response to both TuMV and the aphid vector. They previously found that TuMV was able to reduce callose accumulation [38]. Using arabidopsis mutants and chemical inhibitors of ethylene (ET) synthesis they determined that TuMV can suppress plant resistance to *M. persicae* by altering the ET signalling pathway [45]. This effect was again attributed to NIa-Pro that alters a specific set of ET-responsive defence transcripts. Bak et al. [46] confirmed that ET production promoted aphid attraction to infected plants and virus spread with two potyvirus pathosystems on different host plants (PVY/potato and TuMV/arabidopsis) suggesting that ET signalling may be a general response to potyviruses. 

## 4. Geminiviruses: Master Inventors of Outstanding Pathogenic Factors

*Geminiviridae* represent a family of circular single stranded DNA viruses that are particularly successful in causing devastating diseases in many economically important crops worldwide [47]. They are persistently transmitted by whiteflies, leafhoppers, treehoppers and aphids in a circulative mode [3]. The most devastating and numerous of these viruses are the whitefly-transmitted geminiviruses (genus *Begomovirus*), which have emerged as major pathogens of crops worldwide [47,48,49]. They are acquired from the phloem, translocated from the digestive tract into the haemolymph and then to the salivary glands. The viruses are transmitted via the saliva to a new host plant and can be retained in the insect for their entire life [6]. *Tomato yellow leaf curl virus* (TYLCV) interacts in the insect body with the symbionin protein produced by *Bemisia tabaci* endosymbiont which ensures stability to the virus [50]. Begomoviruses have either monopartite or bipartite genomes. Despite encoding only a few proteins, most begomoviruses are able to manipulate various host components and pathways. In particular, they have evolved diverse RSS targeting plant defence mechanisms and capable of reprogramming multiple pathways such as transcriptional and post-transcriptional gene silencing, hormonal responses, ubiquitin-dependant degradation, autophagy and antiviral chloroplast defence [51,52,53]. Most monopartite begomoviruses associate with betasatellites that enhance accumulation of the helper virus and are essential for induction of typical disease symptoms. Betasatellites share no significant sequence similarity with their helper virus, but depend on them for replication, movement in plants and insect transmission by trans-encapsidation [54]. They encode a single protein (βC1) that determines pathogenicity and symptom phenotype. Numerous studies have shown that βC1 is a key element targeted by plant defence mechanisms and able to repress and manipulate plant defences [54,55]. For instance, βC1 can reverse RNA silencing and interfere with two major protein degradation processes, the ubiquitin-proteasome system and the autophagy pathway [51]. 

Owing to the diversity of host proteins involved in antiviral defence responses and targeted by begomoviruses and their betasatellites, it is conceivable that the RSS and/or the βC1 proteins may indirectly play a role in the feeding behaviour and reproduction of their whitefly vector. Yang et al. [56] addressed the molecular basis of βC1 pathogenic effect encoded by *Tomato yellow leaf curl China Virus* (TYLCCNV). Transgenic arabidopsis expressing βC1 phenocopied the severe disease symptoms naturally observed in virus-infected tobacco plants. The developmental defects were similar to those produced in plants overexpressing AS2, a plant transcription factor that functions together with AS1 to regulate leaf development. Interestingly βC1 was found to specifically interact with AS1, a negative regulator of plant immune response [57] by suppressing the accumulation of miR165/166 which mediate the cleavage of HD-ZIP III transcripts, regulators of leaf polarity. By binding to AS1, βC1 competes with AS2, antagonizes AS1/AS2 dimerization and modulates the expression of several leaf-specific and JA-responsive transcripts (Figure 1, panel C). The betasatellite-induced symptoms could therefore be at least partially explained by the interference of βC1 with leaf morphogenesis. Moreover βC1/AS1 complexes trigger inhibition of several selective JA-responses which may attenuate plant defence response against whiteflies. This work provided the first experimental data in favour of a role of a betasatellite in the repression of the JA-regulated response that could potentially be beneficial to the vector. 

This hypothesis was later confirmed by the work by Zhang et al. [58] in which the betasatellite of TYLCCNV was shown to suppress the JA-regulated defences induced by whiteflies in tobacco and thus improve performances of the insects in infected plants. Similar responses on JA-responsive genes and whitefly population were observed on transgenic tobacco plants expressing the βC1 protein. 

A founder study conducted by Li et al. [59] investigated the molecular basis of the lower resistance to whiteflies observed in plants infected with TYLCCNV and its betasatellite. By analysing the volatiles emitted by TYLCCNV-infected plants, they found that in presence of the satellite, emissions of terpenes were reduced by 75% and this effect was correlated with a significant increase in whitefly attraction. Plant terpenes function either as toxins to inhibit feeding and modify oviposition behaviour, or as repellent to whiteflies; but they can also attract natural predators (parasitoids) [60]. Similar to infected plants, transgenic arabidopsis plants expressing βC1 produced less terpene and were more attractive to whiteflies which laid more eggs. In the same paper Li et al. [59] identified At-MYC2 as the direct target of βC1. As mentioned above MYC2 is a transcription factor and master regulator acting downstream of the JA biosynthesis pathway. *myc-2* mutant lines showed significant reduction of terpene synthase transcripts and were more attractive to whiteflies than wild-type plants. By interacting with MYC2, βC1 also downregulates the glucosinolate-related resistance to whiteflies. Glucosinolates are secondary metabolites that have a defensive function against herbivorous insects [61]. Mechanistically the viral βC1 protein functions as an inhibitor of MYC2 dimerisation to manipulate MYC2-regulated terpene synthase genes, JA signalling and glucosinolate synthesis pathway, to suppress resistance against the whiteflies and promote begomovirus-whitefly mutualism (Figure 1, panel C).

Interestingly in the absence of betasatellite, monopartite begomoviruses have developed other strategies to promote their vectors’ performance. For instance, the TYLCV encodes C2, a known transcription factor for viral genes and a suppressor of gene silencing [51] which can repress specific responses in plant hormone signalling, in particular JA-induced defences and secondary metabolism [62]. Li et al. [63] observed that TYLCV infection improved the survival rate of the whitefly vectors and increased their fecundity. C2-expressing transgenic plants phenocopied these effects and showed repression of specific genes of the JA signalling pathway involved in terpene biosynthesis. In the same paper Li et al. [63] also identified *N. tabacum* RPS27A (a protein with a ubiquitin domain fused to a ribosomal protein) that interacts directly with C2 through its ubiquitin domain. Silencing of RPS27A gene increased the performance of whiteflies which was correlated to inhibition of JAZ1 degradation and repression of genes related to terpene synthesis [63]. Interaction between RPS27A and C2 seems evolutionarily conserved among satellite-less begomoviruses as RPS27A interacts with C2 from another monopartite begomovirus without betasatellite (PaLCuCNV) but not with C2 of TYLCCNV which is associated with a satellite able to promote the performance of whiteflies [64]. 

Several pieces of evidence indicated that βC1 or C2 of different begomoviruses can interfere with several types of post-translational modifications such as ubiquitin-mediated proteasomal degradation, SUMOylation or phosphorylation [49,51]. No experimental data were produced to correlate these effects to a potential virus manipulation of the vector. However as seen before, βC1 encoded by other betasatellites is directly involved in symptom induction and in repression of JA-regulated genes. It is therefore worth mentioning here the following observations that may have a link with vector manipulation. Eini et al. [65] showed that expression of βC1 encoded by Cotton leaf curl Multan virus (CLCuMuV) in transgenic tobacco plants led to a global reduction in levels of polyubiquitinated proteins. CLCuMuV βC1 also interacted with *Nicotiana benthamiana* NbSKP1 [66,67], an essential component of the SCF (SKP1-CUL1-F-box) complex of the E3 ubiquitin ligase family involved in polyubiquitination and subsequent degradation by the 26S proteasome [68] (Figure 1, panel B). SKP1 functions by linking the rest of the complex to an F-box protein, which provides specificity in binding to substrate proteins. Nearly 700 F-box proteins have been predicted, suggesting that plants have the capacity to form a plethora of SCF complexes controlling a multitude of biological processes, among which figure several hormone signalling pathways [68]. For instance, βC1 was shown to increase degradation of SCF^COI1^ (JA receptor) and stabilize GIBBERELLIC ACID INSENSITIVE (GAI) a substrate of the SCF^SYL1^ leading to the repression of JA responses and gibberellic acid (GA) pathways, respectively [67]. Lozano-Duran et al. [69] also demonstrated that *Tomato yellow leaf curl Sardinia virus* (TYLCSV) C2 protein interacts with the catalytic subunit of the CSN (COP9 signalosome) complex, affecting the ability of the CSN to regulate E3 ubiquitin ligase complexes belonging to the SCF family. As a consequence, C2 impairs JA signalling pathway [62,69]. The consequences of JA signalling inhibition on vector behaviour and performance was not addressed in those articles, however it seems very likely that whatever the genome composition (with or without betasatellite), geminiviruses have developed a common strategy to suppress the JA response. Other lines of evidence have shown that geminiviruses interfere with more factors involved in ubiquitin-dependent proteosomal degradation but further link with vector manipulation was not investigated [66]. 

A recent outstanding report explored the case of begomoviruses able to reprogram plant immunity to promote the fitness of their whitefly vector and simultaneously reduce performance of two non-vector herbivores [70]. They showed that the βC1 proteins encoded by the satellites associated to Cotton leaf curl Multan virus (CLCuMuV) and to TYLCCNV can interact with the transcription factor WRKY20 and thereby induce a plant tissue-specific response against different herbivores. In arabidopsis WRKY20 is expressed in vascular tissue and regulates glucosinolate (GS) biosynthesis and other defensive compounds. By interacting with WRKY20, βC1 represses indolic GS biosynthesis in vascular tissues to benefit both the whitefly and the begomovirus while it induces aliphatic GS accumulation in non-vein organs to deter cotton bollworm, a non-vector leaf chewing herbivore. Furthermore, in non-stressed conditions WRKY20 represses SA signalling which is alleviated by the interaction with βC1. This leads to the activation of SA biosynthesis and signalling pathway. Deeper investigations are still required to gain a better understanding of these observations.

## 5. Tospovirus: One More Virus Playing with JA Signalling

The genus *Tospovirus* is the only plant-infecting genus in the *Bunyaviridae* family, the other genera infect vertebrates [71]. Tospoviruses have negative-stranded RNA, protected in enveloped virions. Tospoviruses are exclusively transmitted by thrips and the type member is *Tomato spotted wilt virus* (TSWV), efficiently transmitted by *Frankliniella occidentalis.* Both virus and vector have a particularly wide host range and a worldwide distribution, which makes this pathosystem extremely damaging for many economically important crops [71]. Transmission occurs in a persistent and propagative manner and replication within the vector is essential for high transmission. TSWV is acquired at the larvae stage but only adults can transmit the virus. From the midgut entry site, the virus moves to muscle cells and finally reaches salivary glands [71]. TSWV infection is known to influence behaviour, feeding and fecundity of the vector [72,73] and induce changes in the plant hormonal balance conducive to thrips performance [74]. Interestingly, by switching the host plant during vector development, adult thrips that were exposed to TSWV in their larvae stage showed a preference for healthy plants while the non-exposed thrips preferred the infected leaves [15]. This experiment is clearly a case of direct vector manipulation.

In a recent study Wu et al. [75] investigated the molecular mechanisms underlying TSWV manipulation of the vector behaviour. TSWV-infected pepper or *N. benthamiana* were more attractive to thrips than healthy plants. At the molecular level, monoterpene synthase genes were induced by thrips infestation but repressed when plants were infected by TSWV. Conversely to non-infected plants, virus-infected-plants triggered a reduction in linalool emission which was correlated to the thrips preference. By producing individual TSWV proteins in peppers using a heterologous *Potato virus X*-expression system, NSs was found responsible for the change of the insect behaviour. NSs is a recognized RSS, functioning in both plants and insects, but its mode of action is not yet fully understood [76]. Wu et al. [75] further discovered that NSs interacts directly with MYC2 (Figure 1, panel C), but also with MYC3 and MYC4, three JA-responsive transcription factors. This finding correlates with the thrips’s clear preference for arabidopsis triple KO mutant *myc2/3/4*. This suggests that through its binding to MYC2, NSs could suppress the host defence mediated by MYC2, a core regulator of the JA signalling cascade. Therefore, Wu et al. [75] proposed that TSWV NSs represses JA signalling to attract thrips, increases their performance and thereby improves viral transmission and survival. This report provides an additional example of a viral pathogenicity factor that manipulates the JA signalling pathway induced by the herbivore vector (here thrips) to improve the plant traits for the vector’s benefit and finally for the virus’ propagation.

## 6. *Luteoviridae*: Still a Puzzling Situation

Members of the *Luteoviridae* family are persistent, circulative, non-propagative and phloem-limited viruses [77]. Strictly transmitted by aphids, they are acquired by the vector during extended feeding on the phloem vessels. Their journey in the vector starts by crossing the gut barrier into the hemocoel to reach the salivary glands by successive endocytosis events, before being transmitted to another host plant [4,77]. As mentioned above and conversely to non-persistent viruses, circulative viruses can remain viruliferous for very long periods. A large body of evidence has accumulated showing that viruses from the genus luteovirus and polerovirus alter the quality of their host plants, which then affects the aphid’s host preference, feeding behaviour and performance (reviewed in [9,10,13,18,78]). Infected plants become more attractive to aphid vectors than healthy plants and the aphid vectors often develop more rapidly and produce more offspring on infected hosts. This effect is qualified as an indirect vector manipulation by the virus through the host plant. Preferential colonisation of virus-infected plants by winged aphids was attributed to the yellowing of infected leaves, which are visually more attractive to aphids. However, apterous aphids settle preferentially on virus-infected plants, even in the absence of visual cues, indicating a prominent role of volatile emissions from infected plants that differ from non-infected plants [79,80]. Several studies have highlighted the difference in plant volatile organic compounds (VOC) composition in infected and non-infected plants, those induced by virus-infected plants being more attractive to vectors [81]. Interestingly, the VOC profiles also change with the development of viral infection which might account for the dynamic response of the aphids [82]. A few studies on luteovirids have also reported a direct role of the virus on the vector. Aphids carrying luteovirids showed a clear preference for non-infected plants while non-viruliferous aphids were rather attracted by infected plants [80,83]. Ingwell et al. [16] reported that after having fed aphids on an artificial diet containing purified virus, aphids preferred to settle on healthy plants. Conversely aphids which had not been in contact with virus were preferentially attracted by infected plants.

The first indications concerning the nature of the luteovirid viral proteins and the hormonal pathways involved in plant-aphid interactions were discovered only recently. Patton et al. [84] reported that increased aphid attraction and fecundity on *Potato leafroll virus* (PLRV)-infected potato plants correlated with a reduction of JA and ethylene signalling compared to uninfected control plants. PLRV is the member type of the *Polerovirus* genus. By using transient expression of individual viral proteins in *N. benthamiana* and phytohormone quantification upon aphid settling, they identified three PLRV proteins that triggered changes in plant-aphid interactions: P0, P1 and P7. P0 (which is encoded by poleroviruses but not by luteoviruses like *Barley yellow dwarf virus*) is the suppressor of RNA silencing which promotes the degradation of ARGONAUTE 1 by targeting the host protein to the autophagy pathway through a ubiquitin-dependant mechanism [85,86,87]. P1 is a replication associated protein containing a protease domain and the viral genome linked protein (VPg). P7 is a small protein with a nucleic acid binding activity [88]. Ectopic expression of both P0 and P1 inhibited aphid induction of SA and JA compared to control plants. In absence of aphids, P0 induced ethylene production and P7 inhibited ET emission compared to control plants. However, the presence of aphids had no effect. Deciphering the complex crosstalk of these hormonal pathways upon luteoviral infection will require additional investigations. 

## 7. Perspectives

The five examples of viruses described in this review (cucumovirus, potyvirus, geminivirus, tospovirus and luteovirids) are transmitted by different insect vectors and cover through almost all transmission modes (persistent or non-persistent, propagative or non-propagative). They illustrate a potential common mode of action of viruses that aims to reduce the main plant defences triggered by the insects in the plants and consequently makes the plants more attractive for insects and improves their performance. All viral gene products described here repress the host plant defences induced by the herbivore vector attack and most of them target the JA-signalling pathway. This is not surprising as to promote a beneficial effect for vector-transmitted viruses, viruses have to inhibit the most deleterious plant defence system for the insects. Interestingly the viral proteins target the two main control centres of this pathway, JAZ and MYC2. JAZ degradation by the ubiquitin-dependant proteasome is impaired at the early stage of the pathway and MYC2 transcription activation of JA-regulated genes is blocked further downstream (Figure 1). However other plant hormones are also involved in the plant stress/defence response like SA and ET [32,89]. ET signalling has been shown to mediate aphid attraction to plants infected with potyviruses [45,46]. Reactive oxygen species (ROS) production is another strategy that viruses may use to manipulate vector to enhance virus acquisition and transmission. The RSS CMV-2b was shown to alter aphid’s feeding behaviour by increasing concentrations of H_2_O_2_ in infected plant tissues [90]. It will be important in the future to address the role of the cross-talk between JA and the other hormonal pathways in the relationship between viruses and their vectors. 

Most viral factors identified up till now that are able to manipulate the vector and the host plant to ensure viral transmission, are pathogenicity factors and often display also an RNA silencing suppression (RSS) activity. This is the case of CMV-2b, begomovirus βC1 and C2, tospovirus NSs and polerovirus P0. This dual function targeting two types of plant defence is presumably the key to the remarkable success of these particularly important pests that generate extremely high crop losses in agriculture. Undoubtedly more viral factors will be discovered in the near future and some potential candidates have been identified [91]. Interestingly in the case of potyvirus TuMV, the viral factor that displays activity to promote aphid’s attraction and reproduction is not a known RSS. Westwood et al. [92] showed that Potato virus Y (PVY) and its RSS HC-Pro both repressed the induction of JA-dependant genes. HC-Pro expressed in transgenic plants also increased aphid colony growth rate, a feature that was not observed in PVY-infected plants. Westwood et al. [92] suggested therefore that (i) host-aphid interactions rely on more than just JA-regulated genes (already discussed above) and that (ii) more than one viral gene product is required to mediate plant and vector manipulation. This latter point could even be expanded to the idea that a single protein may have several targets, and viral proteins are indeed often multifunctional. In addition, by targeting one specific host factor that plays an essential role in regulatory functions, the virus could “kill two birds with one stone”. This is the case of the geminivirus RSSs βC1 that binds the MYC2 transcription factor controlling several downstream genes among which the terpene synthase genes control the emitted volatiles and the GS which are repellent secondary metabolites for insects. By targeting SKP1 protein, a common factor to many E3 ubiquitin ligases, βC1 can affect several pathways dependant on proteasome degradation. An essential issue is how the specific target genes are selected in order not to ruin the host plant homeostasis. This level of regulation remains a black box that requires more investigations. 

However, an interesting clue comes from Zhao et al. [70] who showed that the βC1 encoded by the satellites of two begomoviruses regulates a single gene expression in a cell-specific manner by interfering with the interactions between the main regulator WRKY20 and other cellular factors. WRKY20 is a vascular-specific gene that represses the biosynthesis of indole GS in the tissue where whiteflies feed. Conversely aliphatic GS produced in non-vascular tissues have a deterrent activity for other insects like the lepidoptera cotton bollworm. Such a tissue and even time-specific expression has previously been reported for a monoterpene synthase gene in *Nicotiana attenuata* which emits the pheromone β-bergamotene terpene [93]. This gene is induced at night in flowers to produce the pheromone and attract the pollinator and expressed during day time in the leaves to synthesize a defence compound against herbivory attack. 

Deciphering specific regulatory defence pathways to promote the viral vector and deter the non-vector competitors presents a real challenge for the future. This task will require to study more sophisticated plant-virus-insect relations where multitrophic interactions with other herbivores occur in the host plant, moreover in an ecological complex environment. These sophisticated interactions which might be beneficial (synergy) or harmful (parasitism) to the tripartite pathosystem of interest (virus-plant-vector), may occur at different stages of host plant and insect development. The knowledge that will be gained will be of high interest to develop alternative methods to control vector-borne diseases.

During the reviewing process of the manuscript, Wu and Ye [94] published a paper in the special issue “Plant Virus Transmission by Vectors” in Viruses. As it covers the role of JA signalling in the interactions among plant, virus and insect vectors, there are redundancies between the two papers.

## Figures and Tables

**Figure 1 viruses-12-00263-f001:**
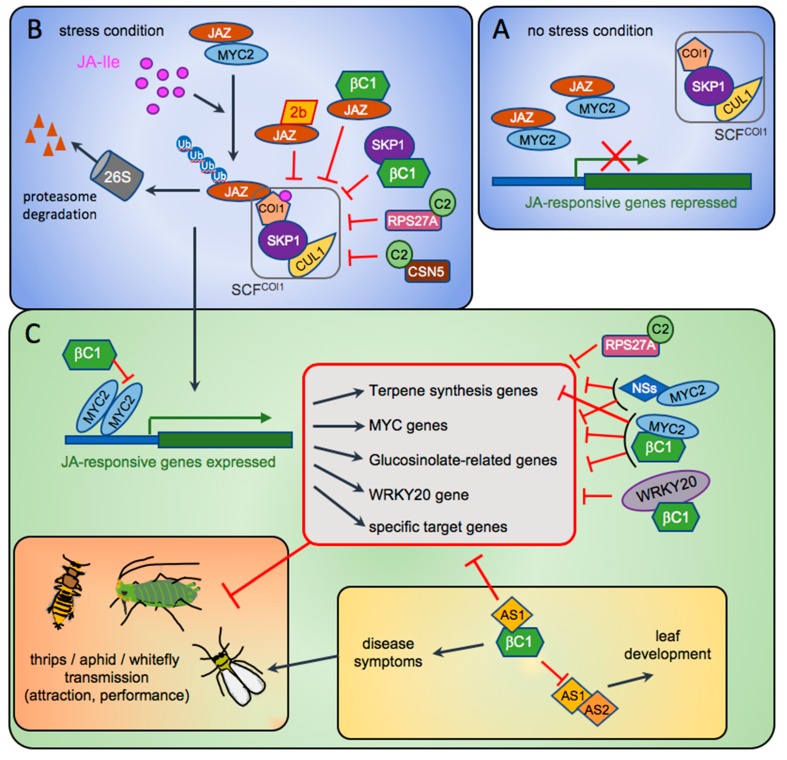
Model depicting the known targets of *Cucumber mosaic virus* (CMV)-2b, *Tomato spotted wilt virus* (TSWV)-NSs and begomovirus C2 and βC1 proteins in the pathways regulating plant defences and symptomatology. (**A**) In non-stressed conditions, the level of active jasmonate, Jasmonoyl-Isoleucine (JA-Ile), is very low. JAZ repressors bind to MYC2 to repress its transcriptional activation on downstream genes of the JA signalling pathway. (**B**) Under herbivore insect attack, the strong increase of JA-Ile triggers the assembly of a JAZ-COI1 co-receptor that will be recruited by the E3-ubiquitin ligase SCF^COI1^ (SKP1/Cullin/F-box). JAZ repressors will undergo ubiquitination and be addressed to the 26S proteasome for degradation. (**C**) The transcription factor MYC2 and others are released and can activate the genes involved in JA response, like those in terpene biosynthesis, glucosinolate production, and other responses (grouped in the red box). During viral infection, viral suppressors of RNA silencing are produced and can antagonise some of these responses. Here are shown four of them, the CMV-2b (panel B), the begomovirus C2, the begomovirus satellite βC1 (panels B and C) and the tospovirus NSs (panel C), interacting with their known targets compromising the SCF^COI1^ complex activity (panel B) or some specific gene activations (panel C). Red bar-heads indicate the inhibitory effect of the viral proteins. The regulations between the various host genes are not reported.

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
