# Peer review of "Molecular Insights into Host and Vector Manipulation by Plant Viruses"

_viruses, 2020, doi:10.3390/v12030263_

Round 1
Reviewer 1 Report
The manuscript by Ziegler-Graff describes the interplay between viral and host factors at molecular level. The manuscript is comprehensive and well drafted. I only have a minor comment that may be considered
The manuscript uses a style "The same authors identified ..." or "...the authors hypothesized...". at multiple places. I recommend writing the authors' names instead. For example "Howe et al hypothesized ..."
Author Response
I would like to thank reviewer 1 for the time he/she took to read and improve the MS.
According to his/her recommendations, I changed all items stating: the authors by their names (lines 120, 331, 504).
Reviewer 2 Report
Dear Author
I found the manuscript informative as it can add up-to-date knowledge on the Plant-virus-vector relationship to the reader. Please go through the attached PDF version of the MS and address the individual points and corrections as specified.

Author Response
I would like to thank the reviewer for the time he/she took to read and improve the MS.
I went through all points and corrections that reviewer 2 highlighted. They are edited in a track changes mode.
In addition, I slightly modified figure 1 by adding one more step in panel B to indicate the initial JAZ-MYC2 complex before JAZ gets degraded. I also added “stress condition” to match to the “no stress condition” (present in panel A). The legend to the figure has been changed at the end to clarify how and where the four viral proteins interact, indicating in which panel to find them. The signification of the red bar-heads has been added.
At several places in the MS, I also added the panel number which is concerned in the relevant sentence to help the reader to find his/her way in the figure. See for example in lanes 133, 135, 322.
The title of the Geminivirus part (4.) that the reviewer seemed not to appreciate, has been modified (lane 249).
Original title: Geminiviruses : a mine of ingenuity to produce diverse outstanding pathogenic factors
The reviewer suggested: augmented ingenuity…
I propose: Geminiviruses : master inventors of outstanding pathogenic factors
I hope this structure is more appropriate.
A few errrors in the references have been corrected.
Reviewer 3 Report
Many complex associations between plant, viruses and their associated vectors causing plant diseases has been demonstrated in the last decade. Based on five selected examples that cover most of the insect transmission modes, this review focus on the mechanisms underlying virus/plant interactions that modify the plant host metabolisms and the vector behaviours in order to improve the transmission process.
I ‘ve really appreciate the reading of this review. This manuscript is well-written, well-structured, and documented with all the recent data. The idea to described the interplay between viral and plant factors that lead to an efficient transmission, through five examples make the reading easy and very comprehensive. The section headlines are relevant of the content of each section and the figure propose a good view of the overall molecular interaction during the transmission process. This review clearly demonstrate that the author has a good knowledge of the research area.
I don’t really think that my comments could improve this study further. My recommendation is to publish this manuscript.
Author Response
I would like to thank reviewer 3 for the time he/she took to read and improve the MS.